# Dielectric Function and Magnetic Moment of Silicon Carbide Containing Silicon Vacancies

**DOI:** 10.3390/ma15134653

**Published:** 2022-07-01

**Authors:** Sergey A. Kukushkin, Andrey V. Osipov

**Affiliations:** Institute for Problems in Mechanical Engineering of the Russian Academy of Sciences, 199178 Saint-Petersburg, Russia; andrey.v.osipov@gmail.com

**Keywords:** silicon carbide, spin polarization, dielectric function, half-metallic ferromagnet, density-functional theory

## Abstract

In this work, silicon carbide layers containing silicon vacancies are grown by the Method of Coordinated Substitution of Atoms (MCSA). The main idea of this fundamentally new method is that silicon vacancies are first created in silicon, which is much simpler, and only then is silicon converted into silicon carbide by chemical reaction with carbon monoxide. The dielectric function of silicon carbide containing silicon vacancies, grown on both n- and p-type silicon substrates, is measured for the first time. The density functional method in the spin-polarized approximation is used to calculate the dielectric function of silicon carbide containing silicon vacancies. It is shown that the influence of the magnetic moment of vacancies on the dielectric function is decisive. Qualitative correspondence of the computational model to the obtained experimental data is demonstrated. It is discovered that silicon vacancies make silicon carbide much less transparent. It is shown that the imaginary part of the dielectric function is described as a sum of oscillatory peaks in the form of the Gaussian functions. Vacancies lead, as a rule, to one or two additional peaks. According to the amplitude and position of the additional peaks, it is possible to qualitatively estimate the concentration of vacancies and their charge.

## 1. Introduction

In recent years, a great interest has begun to appear in the development of semiconductor devices that use for their work the spin of charge carriers and not the charge itself [1], including spin-polarized light-emitting diodes, spin field-effect transistors, single-photon detectors, single-photon emitters, and quantum bits. Therefore, the development of new semiconductor materials with a magnetic moment and the property of spin polarization is a very relevant task [1,2,3]. Moreover, for scaling and providing the interconnection between the elements, it is desirable that these materials combine well with the available materials of microelectronics, such as silicon Si, silicon carbide SiC, and semiconductor compounds A^3^B^5^ [4].

Previously, the magnetic properties of materials were associated only with d- and f-orbitals of electrons in transition or rare-earth elements. Thus, magnetic semiconductors used to be produced by doping optically transparent semiconductors with transition metals. Eventually, it has become clear that even more efficient spintronic properties can be obtained with p-electrons by using point defects in semiconductors [3,4]. In particular, the most studied solid-state spin is the negatively charged nitrogen-vacancy center ($NV^{-}$) in diamond [3]. Recently, similar NV centers in SiC have started to be actively studied [5]. Studies have shown that silicon vacancies $V_{Si}$ in silicon carbide also provide effective spintronic properties [6]. That is why SiC with various point defects based on $V_{Si}$ is now considered as one of the most promising materials for spintronics [1,2,5,6].

It is quite difficult to obtain silicon vacancies $V_{Si}$ in SiC, because the energy of their formation is very high; in particular, it is approximately equal to 8 eV in cubic 3C–SiC [7]. Therefore, to create them, high-energy beams of electrons, as well as ions, are usually used [8]. As not only $V_{Si}$ but also other defects are formed in this case, the quality of SiC with $V_{Si}$ is not very high upon irradiation. Recently, in the work [9], a much simpler and more reliable method for obtaining silicon vacancies in SiC was proposed and implemented, based on the fact that the formation energy of $V_{Si}$ in silicon Si is much lower (~3.3 eV) than in SiC. Initially, vacancies are created in Si thermally (i.e., by heating in vacuum to a temperature of T~1340÷1380 °C). Then, Si is converted into SiC by the method of coordinated substitution of atoms (MCSA) by the chemical reaction [10,11,12,13]
(1)2Si(crystal)+CO(gas)=SiC(crystal)+SiO(gas)↑.

Therewith, some vacancies in Si simply transform into silicon vacancies in SiC [13]. Another part of $V_{Si}$ disappears when Si collapses into SiC with a twofold decrease in the volume of the material [10,11]. The term “coordinated” means that new SiC chemical bonds are created simultaneously with the destruction of former Si-Si chemical bonds in the process of the chemical reaction [10,11]. The overall diamond-like bond structure does not change in this case, which makes it possible to obtain high-quality layers of cubic 3C–SiC containing $V_{Si}$ by this method. The concentration of $V_{Si}$ in SiC is determined by the time of preliminary annealing of Si in vacuum [9,12,13]. The longer the annealing, the greater the amount of $V_{Si}$.

In the review [10], a sequential thermodynamic and kinetic theory of Si transformation into SiC in the course of chemical reaction (1) was constructed. It was shown that the process of transformation of Si into SiC consists of two stages. In the first stage of reaction (1), the CO molecule interacts with the surface of the silicon substrate. The oxygen atom reacts chemically with the Si atom to produce SiO gas. This reaction proceeds through the formation of an intermediate Si–O–C complex forming an almost equilateral triangle [14]. In fact, the oxygen atom plays the role of a reaction catalyst, without which reaction (1) is impossible. Then, the SiO gas is removed from the system, and a silicon vacancy $V_{si}$ is formed in place of the substrate silicon atom that turned into SiO gas. As this is the first kind of phase transition, it occurs with the formation of a new phase nucleation. In this case, this nucleus is a complex formation consisting of the SiC itself, the pore under its surface, and the surrounding shrinkage pores. An image of such a nucleus is given in the review [11]. For the thermodynamic and kinetic description of this kind of nucleation, the equations of the standard nucleation theory had to be significantly modified. The derivation and solution of these equations are given in the review [10]. In the same work, a theory of the formation of such nuclei on vicinal surfaces of Si crystals deviated by a small angle from planes (111) was also constructed.

The Density Functional Theory (DFT) studies [7,15,16] show that many properties of SiC containing $V_{Si}$, including the electric charge localized near $V_{Si}$, are determined by the position of the Fermi level. This is especially important for SiC layers grown on a Si substrate by MCSA, as the substrate can be doped with atoms of boron B, as well as phosphorus P or antimony Sb. In the first case, the Fermi level of Si shifts to the bottom of the valence band; in the second case, it shifts to the top of the conduction band. The DFT calculations show that $V_{Si}$ in SiC contacting doped Si acquires an electrical charge. If Si is doped with B, then $V_{Si}$ receives a positive charge of either +e or +2e depending on the position of the Fermi level, i.e., concentration of B in Si [15]. As a rule, the concentration of boron in silicon is low and provides a conductivity of the order of $50÷1000\;{\left(O\;\mathrm{hm}\cdot \mathrm{cm}\right)}^{-1}$, so the case of +e is more likely. If Si is doped with P or Sb, then $V_{Si}$ receives an opposite charge of either −e or −2e, also depending on the position of the Fermi level [7,15,16]. It is very important to point out that if SiC is grown from Si doped with B, then at temperatures >1200 °C, a carbon atom C closest to $V_{Si}$ leaps to the place of $V_{Si}$, which leads to a significant decrease in the total energy [15]. Therewith, four carbon atoms form an almost flat cluster, while a void with a diameter of 2.1 Å is formed under it at a distance of 1.8 Å [9]. Without a charge or with a charge of +e, this formation has a magnetic moment equal to the Bohr magneton $\mu_{B}$ due to the unpaired p-electron of the C atom located in the center of the cluster at the site of $V_{Si}$ [9]. As the magnetic and electrical properties of such a system are very similar to the $NV^{-}$ center in diamond, this formation is named the C_4_V center in the work [9]. If Si is doped with P or Sb, then such a leap of the C atom to the place of $V_{Si}$ is energetically unfavorable due to the received negative charge. Studies carried out in this work have shown that SiC has an even larger magnetic moment in this case, which leads to a spin polarization of SiC, which, in turn, affects the dielectric function and many other properties of SiC. Experimental data show that the dielectric function of SiC grown by the MCSA always differs from the dielectric function of SiC grown, for example, by the modified Lely method [17]. SiC grown by MCSA is always less transparent in the range of 0.5÷5 eV; moreover, an increased absorption of light is often observed in the range of 1.0÷2.5 eV. Notably, small differences in the conditions of the SiC growth can lead to significant differences in the dielectric function. From the results of [9], it becomes clear that the reason for this is that SiC obtained by MCSA always contains $V_{Si}$ that were originally in Si, while SiC obtained by any other method does not contain $V_{Si}$.

Up to the present time, all the features of the measured dependences of the dielectric function of SiC on the photon energy have been explained using Bruggeman’s model of the effective-medium approximation (EMA), which mixes SiC with crystalline carbon and pores [18]. This is because the dielectric function of crystalline carbon successfully exhibits the necessary behavioral features at both low and high photon energies. At low concentrations of $V_{Si}$ (up to about 1%), the EMA model works quite well, but with an increase in the time of pre-annealing of Si in vacuum to 1 min or more at a temperature of 1350 °C or more (which leads to an increase in the concentration of $V_{Si}$ up to several percent), the EMA model ceases working, especially at low photon energies of 0.5÷2.5 eV. Therefore, the purpose of this work is to study the dependence of the dielectric function of SiC containing $V_{Si}$ on the photon energy by the DFT method and compare the obtained theoretical results with the experimental ones. As all the obtained results are determined by the magnetic moment of SiC, the band structure of SiC is also determined according to spin.

## 2. Materials, Methods and Measurement Results

To measure the dielectric function of 3C–SiC containing silicon vacancies at different positions of the Fermi level, two samples with different concentrations of defects were grown by the MCSA. One sample was grown on a Si(111) substrate doped with Sb (the electrical conductivity of the substrate material was $\sim 5\;{\left(\mathrm{Ohm}\cdot \mathrm{cm}\right)}^{-1}$) and the second sample was grown on a Si(111) substrate doped with B (the conductivity of the substrate material was $\sim 50\;{\left(\mathrm{Ohm}\cdot \mathrm{cm}\right)}^{-1}$). Before the synthesis, both substrates had been cleaned of oxides in a mixture of NH_4_OH and NH_4_F, which additionally ensured the passivation of the Si(111) surface by hydrogen atoms [19]. The substrates were then annealed in vacuum to create vacancies, and then 3C–SiC layers were grown on the substrates by the MCSA using the chemical reaction with CO (1). For the Si(111) substrate doped with Sb, the pre-annealing time was 0.5 min, the growth time in CO atmosphere at the pressure $p_{CO}=300\;\mathrm{Pa}$ was 5 min, and the temperature of annealing and growth was 1340 °C. For the Si(111) substrate doped with B, the pre-annealing time was 5 min, the growth time in CO atmosphere at a pressure of 300 Pa was 15 min, and the temperature of annealing and growth was 1380 °C, which provided a higher concentration of vacancies compared to the first case. The produced 3C–SiC layers with silicon vacancies were examined in the photon energy range of 0.5÷9.3 eV with the J.A. Woollam VUV-VASE ultraviolet ellipsometer with a rotating analyzer.

The measured dependence of the pseudodielectric function ε on the photon energy E for the first sample, i.e., for the n-type substrate (Si doped with Sb), is shown in Figure 1a. To analyze this dependence, we used the previously proposed two-layer ellipsometric model [18] consisting of a substrate containing Si, SiC and pores, a buffer layer with a thickness of ~2 nm (layer No. 1), a SiC layer (layer No. 2), and roughness on its surface. Approximating the imaginary and real parts of the dielectric function of a SiC layer in the form of splines, one can extract them from the experimental data. Minimization of the difference between the theoretical and experimental dependences ε(E) with respect to given parameters [20] makes it possible to determine the thickness of a SiC film and the dielectric function of SiC. In the present case, the thickness of the SiC layer with silicon vacancies turned out to be 120 nm (with the contributions of the buffer layer and roughness). The dielectric function of SiC with silicon vacancies for this sample is shown in Figure 1b. In accordance with the classical theory of light dispersion [20], the imaginary part of the dielectric function can be represented as a sum of oscillators (there are 4 of them in the present case). Their shape is best described by the Gaussian function:(2)$$\varepsilon_{2}\left(E\right)=\overset{n}{\underset{i=1}{\sum }}A_{i}e^{-{\left(\frac{E-E_{i}}{\sigma_{i}}\right)}^{2}}$$

The energies of these oscillators are 0.75 eV, 5.95 eV, 6.95 eV, and 9.2 eV and are marked by dashed lines in Figure 1b. The real part of the dielectric function $\varepsilon_{1}\left(E\right)$ was calculated using the Kramers–Kronig relation [17,20]. The measured dependence $\varepsilon_{2}\left(E\right)$ of 3C–SiC containing silicon vacancies differs from $\varepsilon_{2}\left(E\right)$ of an ideal 3C–SiC primarily in the appearance of a new oscillatory peak with an energy of 0.75 eV, which makes SiC less transparent. Another important difference is that the energies of the main SiC peaks decrease by about 0.3 eV, which also makes SiC less transparent. In addition, the amplitudes of the main peaks decrease.

The measured dependence of the pseudodielectric function ε on the photon energy E for the second sample, i.e., for the p-type substrate (Si doped with B), is shown in Figure 2a. To analyze the obtained dependence, the previously proposed two-layer ellipsometric model [18] was used as well. The thickness of the SiC layer in this case is 480 nm, meaning that vacancies strongly stimulate the synthesis of 3C–SiC. In both cases, the film thicknesses obtained by ellipsometry coincide with those observed by scanning electron microscopy (SEM) (see Figure 3). Without the pre-annealing step, the thickness of the SiC layer would have been about 70–100 nm under the same growth conditions. The imaginary part of the dielectric function of SiC with silicon vacancies for this sample is shown in Figure 2b. It is represented as a sum of 5 oscillators having the form of the Gaussian function (2). The energies of these oscillators are 1.6 eV, 2.6 eV, 6.2 eV, 7.6 eV, and 9.6 eV and are marked by dashed lines in Figure 2b. In this case, already two new oscillatory peaks with energies of 1.6 eV and 2.6 eV appear, making SiC much less transparent. This is what leads to a very strong decrease in the amplitude of the interference peaks in the ellipsometric spectrum in the range of 2–5 eV (Figure 2a), as the intensity of the light reflected from the bottom of the SiC layer is now significantly lower in this range of photon energies. An even greater decrease in the amplitudes of the three main oscillatory peaks should also be noted (Figure 2b).

The XRD (X-ray diffraction) spectra of two 3C–SiC/Si(111) samples containing silicon vacancies are shown in Figure 4. Both spectra are very similar to each other. It can be seen that there are only peaks corresponding to the <111> direction, which indicates the crystalline perfection of 3C–SiC in both cases. Figure 5 shows the RHEED (reflected high-energy electron diffraction) patterns of the two samples after cooling, obtained on the EMR-100 electron diffractometer with an electron energy of 50 keV. At such an energy of electrons, the depth of their penetration into the sample does not exceed ~ 100 nm; therefore, this diffraction pattern corresponds only to the upper layer of epitaxial SiC. It can be seen that there is only a diffraction pattern corresponding to a smooth epitaxial layer. Patterns corresponding to a polycrystalline or amorphous structure are completely absent. This also indicates the epitaxial quality of 3C–SiC samples containing silicon vacancies for both the n-type and p-type substrates.

## 3. Modeling Results

In this work, to model the properties of SiC containing $V_{Si}$, we used the DFT methods implemented in the MedeA VASP software [21]. Periodic boundary conditions in all three dimensions, as well as the spin-polarized approach [22], were used in all the calculations, which makes it possible to adequately describe the magnetic moment in SiC. In calculating the energy of the system and optimizing its geometry, the exchange-correlation contribution was computed using the PBE-GGA functional [23], which is better than the others for calculating the magnetic moment in such systems. The exchange-correlation interaction in calculations of the dielectric function and band structure was calculated using the mBJ-LDA meta-GGA functional [24], which provides high accuracy in calculating the bandgap (the error is ~1% for pure 3C–SiC), as well as acceptable accuracy in calculating the magnetic moment. For integrating over the Brillouin zone, we used a grid of k-points generated according to the Monkhorst–Pack scheme, where the distance between them was no more than $0.25\;{Å}^{-1}$. In all the calculations, the projector augmented-wave (PAW) method [21] was used, and the cutoff energy of the plane waves was 400 eV.

First, it is necessary to find the minimum-energy configuration of atoms of the cubic 3C–SiC polytype (as it is this polytype that grows from Si by the MCSA) containing $V_{Si}$. The key question here is in which cases is it beneficial for the C atom closest to $V_{Si}$ to leap to the place of $V_{Si}$, and in which cases is it not. The silicon vacancy $V_{Si}$ in 3C–SiC is shown in Figure 6a as a void with a diameter of 3.0 Å. Four C atoms that had been bonded with the removed Si atom are marked in blue there. The p-electrons of just these atoms provide SiC with a magnetic moment. Following the work [9], we call the point defect formed after the leap of the C atom to the place of the silicon vacancy as the $C_{4}V$ center by analogy with the NV center in diamond. The $C_{4}V$ center in 3C–SiC, consisting of an almost flat cluster of four C atoms with a bond length of 1.56 Å and a void with a diameter of 2.1 Å located at a distance of 1.8 Å below the cluster, is shown in Figure 6b. The central atom of the cluster, the p-electrons of which provide a magnetic moment of SiC, is marked in blue.

For these two systems $C_{4}V$ and $V_{Si}$, the configurations of atoms corresponding to the minimum energies $E_{C_{4}V}$ and $E_{V_{Si}}$ of the systems without additional charge, as well as in the presence of localized charges +e, −e, and −2e, were calculated by the DFT method. Along with that, the magnetic moment μ of a system and the magnetic energy $E_{M}$, i.e., the difference in the energy of a system without a magnetic field and with a magnetic field (the energy is always lower in the latter case), were calculated. The results of the calculations for a 3C–SiC supercell consisting of 47 Si atoms and 48 C atoms and having a size of $12.33\times 12.33\times 7.55\;{Å}^{3}$ with the P3m1 symmetry (Figure 6) are presented in Table 1 (the leap of the C atom does not change the P3m1 symmetry). The energy $E_{V_{Si}}$ of the system with the vacancy was taken as a reference point to calculate the value of $E_{C_{4}V}$; the magnetic moment is expressed in units of the Bohr magneton $\mu_{B}$.

From the data given in Table 1, it can be seen that the $C_{4}V$ configuration is most energetically favorable in the absence of charge, as well as at q=+e, whereas the $V_{Si}$ configuration is more favorable at q=−e, q=−2e. In all cases, the most favorable configuration has a magnetic moment. The largest magnetic moment$\;\mu =3\mu_{B}$ and the largest magnetic energy 0.46 eV correspond to the silicon vacancy $V_{Si}$ in the presence of the charge −e. In this case, only the magnetic moment keeps the C atom from leaping into $V_{Si}$ because $E_{M,V_{Si}}>E_{C_{4}V}$. The above calculations describe the leap of the C atom under the condition that the charge q localized near the defect is conserved. However, in the general case, the leap can be accompanied by a change in the charge. In this case, it is necessary to calculate the configuration energy of each system for all q in dependence on the position of the Fermi level, as the energy of an electron (or hole) localized near the defect is equal to the Fermi energy, and then to compare all energies with each other, choosing the minimum one. The results of calculations performed in various approximations and for various SiC polytypes [7,15,16] generally agree with the results of the above calculations for 3C–SiC and constant q. Namely, the case with no charge q=0 is either never realized, or it happens in a very narrow range of the Fermi level. If SiC has the p-type conductivity, or SiC is in contact with Si of the p-type conductivity (i.e., doped with boron B), then the case of q=+e or q=+2e is realized. In all these cases, the C atom leaps to the site of $V_{Si}$ and the $C_{4}V$ center is formed. If the level of doping with B is not very high, then the $C_{4}V$ centers donate one electron (possibly, two electrons sometimes) to B atoms located in the Si sites of 3C–SiC or Si, as it is energetically favorable. In this case, the magnetic moment of the each $C_{4}V$ center with q=+e (as well as q=0) is equal to the Bohr magneton. The difference in the density of electrons with spin “up” and “down” is shown in Figure 7. The semitransparent blue surface corresponds to a difference of $0.05\;e/{Å}^{3}$. It can be seen that in the case of the $C_{4}V$ center, the main contribution to the magnetic moment is made by the unpaired p-electron of the C atom located in the center of the cluster and therefore having only 3 bonds (Figure 7b). The p-electrons of three Si atoms with dangling bonds also make a small contribution. In the case of the vacancy $V_{Si}$, the main contribution to the magnetic moment comes from the unpaired p-electrons of four C atoms with dangling bonds (Figure 7a).

In order to investigate the dependence of the dielectric constant of SiC on the concentration of silicon vacancies, three atomic configurations of different sizes have been studied. The first configuration consists of 26 Si atoms and 27 C atoms and has cell sizes a=b=9.25 Å and c=7.55 Å; the second system consists of 47 Si atoms and 48 C atoms (a=b=12.33 Å and c=7.55 Å) (see Figure 6 and Figure 7); the third system consists of 95 Si atoms and 96 C atoms (a=b=12.33 Å and c=15.10 Å). In all the cases, α=β=90° and γ=120°, and the symmetry corresponds to the trigonal group P3m1. These three systems correspond to three concentrations of silicon vacancies $n_{V_{si}}=1/27\approx 3.7\%,$ $n_{V_{si}}=1/48\approx 2.1\%$, and $n_{V_{si}}=1/96\approx 1.04\%$, respectively. First, for each of the 4 charge values of the system (q=+e, q=0, q=−e, and q=−2e), the atomic configuration corresponding to the minimum of the energy in the spin-polarized approximation, taking into account the magnetic moment of the system, was found. At that, the cases q=+e and q=0 corresponded to the configuration with $C_{4}V$ (Figure 6b), whereas the cases q=−e and q=−2e corresponded to the configuration with $V_{Si}$ (Figure 6a). Then, for each case, the band structure and dielectric function were calculated in the spin-polarized approximation. For this, the mBJ-LDA meta-GGA functional [24], which provides the highest accuracy in the calculation of the bandgap, was used. As this system has a solid-state spin, the behavior of the energy bands with spin “up” is very different from that of the energy bands with spin “down”.

The calculation of the band structure of silicon carbide containing silicon vacancies showed the following. For electrons with spin “up” (to be definite), 3C–SiC remains an indirect-bandgap semiconductor with a bandgap of ~ 2.3 eV, as before. Only the bandgap width slightly decreases with the increase in the concentration of silicon vacancies, whereas for electrons with the opposite spin, i.e., with spin “down”, the band structure changes dramatically. The bandgap decreases very strongly due to unpaired p-electrons of carbon atoms with three bonds (Figure 6 and Figure 7). As a rule, at a low concentration of silicon vacancies $n_{V_{Si}}≲2\%$ and a small charge q=+e, q=0, and q=−e, 3C–SiC remains a semiconductor with a bandgap width of ~ 0.5÷1.0 eV (the exact value of the bandgap width depends on the symmetry group, i.e., on the relative positions of vacancies). With an increase in the concentration $n_{V_{Si}}$ or an increase in charge to q=−2e, 3C–SiC can turn into a magnetic semimetal, i.e., the bandgap can decrease to zero for spin-down electrons. As a typical example, Figure 8 shows the calculated band structure for the $V_{Si}$ system consisting of 26 Si atoms and 27 C atoms with the P3m1 symmetry group at charges q=+e, q=−e, and q=−2e, as well as for the supercell of an ideal 3C–SiC consisting of 27 Si atoms and 27 C atoms (the symmetry group, of course, changes in this case and corresponds to P1). Bands for spin-up electrons are shown in red, and those for spin-down electrons are shown in blue. In particular, at q=−e, the bandgap for spin-up electrons is approximately 2.0 eV, whereas the bandgap for spin-down electrons is almost direct and has a width of 0.9 eV (Figure 8c). This means that spin-down electrons, which are responsible for the appearance of new bands near the Fermi level, will intensely absorb photons with an energy of the order of ~ 1.0 eV. The total number of such electrons is not very large and is proportional to the number of silicon vacancies in 3C–SiC. At q=−2e, this system turns into a magnetic semimetal and absorbs light of about 1 eV even more intensely. At q=+e, an additional band appears for spin-down electrons just below the Fermi level, which also leads to additional absorption of light (Figure 8b). The results of calculating the band structure of systems with q=+e for the P31m and Pm symmetries are given in the work [9].

Thus, the magnetic moment of silicon vacancies in SiC leads to the appearance of additional peaks in the imaginary part $\varepsilon_{2}$ of the dielectric function. The real part $\varepsilon_{1}$ of the dielectric function behaves in accordance with the Kramers–Kronig dispersion relation [17]. Figure 9 shows the results of the calculations of the dielectric function for the above-mentioned systems with the P3m1 symmetry group and different concentrations of silicon vacancies at different q from +1e to −2e.

The general trend is as follows. When silicon vacancies appear, silicon carbide becomes less transparent. For q=+e and q=0, i.e., for the $C_{4}V$ structure, the imaginary part $\varepsilon_{2}$ of the dielectric function, related to the absorption of light, acquires additional peaks in the range of ~ 0.5÷2.5 eV. When the concentration of vacancies is over 2.5÷3%, several peaks may appear. The function ε(E) at q=0 is very close to ε(E) at q=+e and is not shown in Figure 5. At q=−e and q=−2e (i.e., for the $V_{Si}$ structure), $\varepsilon_{2}$ acquires additional peaks in the range of ~ 0.5÷2.5 eV, and their amplitude is somewhat higher than in the case of the $C_{4}V$ structure. When the concentration of vacancies is over 2.5÷3%, several peaks may appear as well, and they can merge together. It should be emphasized that the specific type of function strongly depends not only on the concentration of vacancies and the vacancy charge q, but also on the relative position of vacancies, i.e., symmetry groups of the system. The calculations have shown that the additional peaks of the function $\varepsilon_{2}$ are best described by a sum of the Gaussian functions (2). The function $\varepsilon_{2}\left(E\right)$ for an ideal 3C–SiC that does not contain vacancies, in the range of 0.5÷9.3 eV corresponding to the energy range of the ultraviolet ellipsometer, is described by the sum of 3 Gaussian functions with maxima $E_{1}=6.2\;\mathrm{eV}$, $E_{2}=7.3\;\mathrm{eV}$, and $E_{3}=9.5\;\mathrm{eV}$. The presence of silicon vacancies slightly changes these 3 peaks and leads to an appearance of new peaks, usually one or two. At a concentration of silicon vacancies ≲3%, it is usually sufficient to introduce one additional Gaussian function. We also note that the main peaks can be described by the Tauc–Lorentz functions [17], while the additional peaks are described by the Gaussian functions (2).

## 4. Discussion

Overall, the obtained theoretical results are in complete agreement with the experimental data. The presence of silicon vacancies in 3C–SiC makes this material less transparent and turns it into a magnetic semiconductor. For the spin-up case, the bandgap decreases slightly, approximately by 0.3 eV for a 2% share of silicon vacancies. For the spin-down case, the bandgap decreases significantly, to a value of ~ 1 eV for the vacancy charges q=+e, q=0, and q=−e and to a value of 0÷0.5 eV for the vacancy charge q=−2e. The p-electrons of carbon atoms with dangling bonds are responsible for this (Figure 6). The greater the concentration of silicon vacancies, the greater the number of these electrons and the greater the amplitude of the oscillatory peak (or several peaks) of the $\varepsilon_{2}$ function, associated with the vacancies. The qualitative dependence of the dielectric function ε on the photon energy E can be seen in Figure 9 at various q and concentrations of $V_{Si}$. In particular, with the help of Figure 9, from the amplitude and location of the new peaks, one can qualitatively estimate the concentration of vacancies for the two 3C–SiC/Si(111) samples studied above, namely $n_{V_{Si}}\sim \;0.5÷1.5\%$, q=−2e for the first sample grown on n-type Si (Figure 1b), and $n_{V_{Si}}\sim \;3÷5\%$, q=+e for the second sample grown on p-type Si (Figure 2b). More accurately, the function ε(E) can be calculated by the DFT method if positions of vacancies relative to each other, i.e., symmetry groups of the system, are known. At the same concentration of vacancies, SiC grown on p-type Si is somewhat more transparent than SiC grown on n-type Si, which may be due to the fact that only one carbon atom has dangling bonds in this case, and not three (Figure 6 and Figure 7). Therefore, the density of the spin-down electrons at new levels near the Fermi level is a bit less than in the case of n-type.

## 5. Conclusions

In summary, in this work, the dielectric function of 3C–SiC containing silicon vacancies has been calculated by the DFT method. It has been shown that the dependence of the dielectric function ε on the photon energy E is completely determined by the magnetic properties of silicon vacancies in 3C–SiC, which, in turn, are determined by the doping material in silicon. When silicon is doped with boron, the charge of the vacancy becomes positive, and the carbon atom leaps to the place of the silicon vacancy. The magnetic moment of this $C_{4}V$ structure is equal to $\mu_{B}$ (see Table 1), and the imaginary part of the dielectric function acquires an additional oscillatory peak in the range of 2÷4 eV. With an increase in the concentration of the vacancies, more peaks may appear, and they can merge with each other (Figure 9), because there are more additional energy bands near the Fermi level. When silicon is doped with phosphorus or antimony, the vacancy charge becomes negative, and the magnetic moment reaches up to $3\mu_{B}$ (see Table 1). At q=−e, it is the magnetic field that keeps the carbon atom from leaping to the place of the missing silicon atom. In this case, the imaginary part of the dielectric function also acquires an additional oscillatory peak in the range of 0.5÷2.5 eV. Again, with an increase in the concentration of the vacancies, more peaks may appear, and they can merge with each other (Figure 9). It has been shown that the imaginary part of the dielectric function of 3C–SiC containing silicon vacancies can be represented with high accuracy as a sum of several (mostly 4÷5) peaks in the form of the Gaussian functions (2). An increase in the concentration of vacancies leads to an appearance of additional peaks, a shift to the left of the main peaks, and also to a decrease in the amplitude of the main peaks. The performed experiments have shown a qualitative agreement between the theoretical and experimental results. It has been shown that the pre-annealing of silicon in vacuum within the period of 0.5÷15 min at a temperature of 1340÷1380 °C leads to a concentration of the vacancies in silicon carbide of the order of 0.5÷5%.

Studies performed as a result of this work and earlier work [25] have shown that SiC grown on Si by the MCSA method is a very promising material for use in spintronics and the creation of new types of electronic devices based on it. In this sense, silicon carbide obtained using the method of pre-annealing Si in vacuum is especially interesting, because in this case, SiC is maximally saturated with silicon vacancies, which provide a SiC magnetic moment. A study of the magnetic properties of SiC films on Si performed on SiC obtained by the MCSA method without silicon pre-annealing [25] can serve as a proof of this. This SiC contains a much lower vacancy density than the SiC obtained with the pre-annealing. In addition, nevertheless, strong magnetic susceptibility was experimentally detected in the samples of this silicon carbide [25]. Obviously, in SiC samples with a high vacancy content, this effect will only increase, which we hope will be damaged in the near future by our experiments.

## Figures and Tables

**Figure 1 materials-15-04653-f001:**
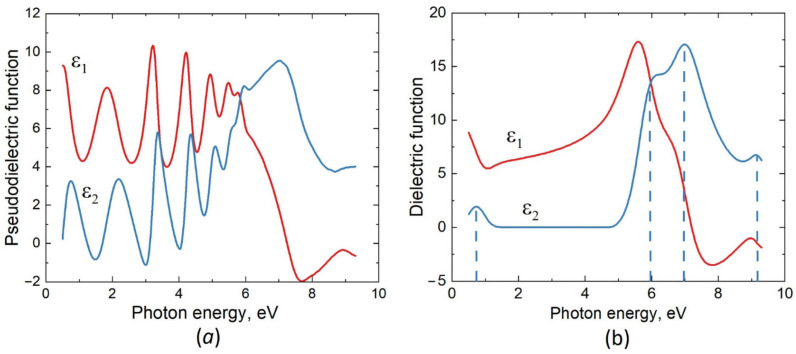
Pseudodielectric function of a 3C–SiC sample grown by MCSA on the n-type Si(111), as measured with the VUV-VASE ellipsometer (**a**), and dielectric function of the sample, as extracted from the measured data (**b**). $\varepsilon_{1}$ is the real part and $\varepsilon_{2}$ is the imaginary part; the dashed lines show the positions of the peaks of the $\varepsilon_{2}$ function.

**Figure 2 materials-15-04653-f002:**
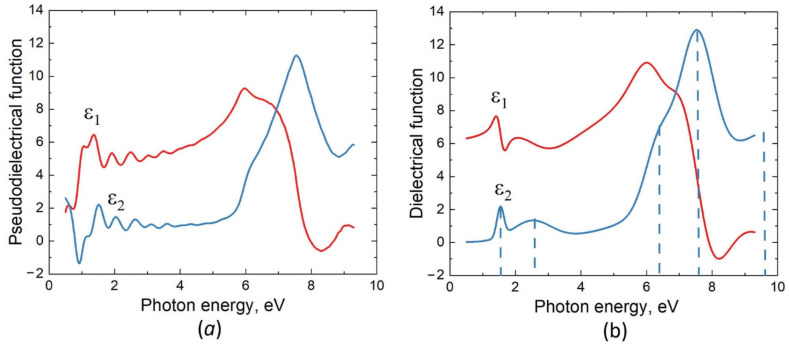
Pseudodielectric function of a 3C–SiC sample grown by MCSA on the p-type Si(111), as measured with the VUV-VASE ellipsometer (**a**), and dielectric function of the sample, as extracted from the measured data (**b**). $\varepsilon_{1}$ is the real part and $\varepsilon_{2}$ is the imaginary part; the dashed lines show the positions of the peaks of the $\varepsilon_{2}$ function.

**Figure 3 materials-15-04653-f003:**
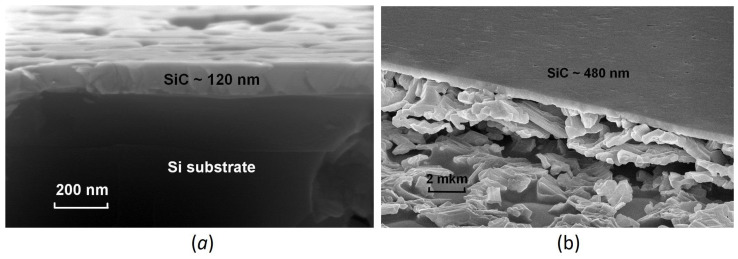
SEM images of 3C–SiC samples containing silicon vacancies grown on Si of n-type (**a**) and p-type (**b**).

**Figure 4 materials-15-04653-f004:**
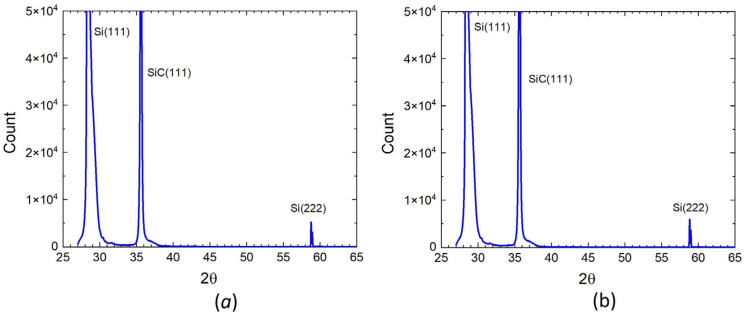
XRD spectra of 3C–SiC samples containing silicon vacancies grown on Si of n-type (**a**) and p-type (**b**).

**Figure 5 materials-15-04653-f005:**
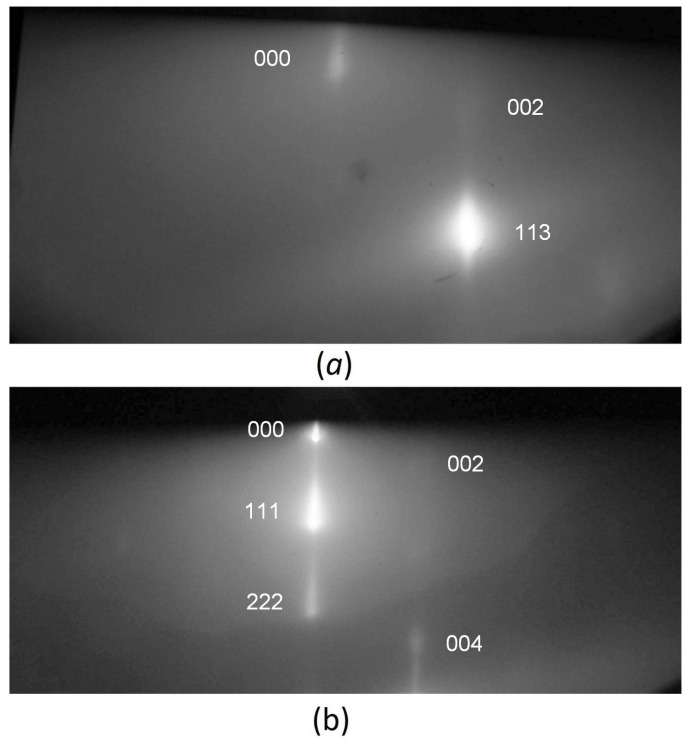
RHEED patterns of 3C–SiC samples containing silicon vacancies, grown on Si of n-type (**a**) and p-type (**b**).

**Figure 6 materials-15-04653-f006:**
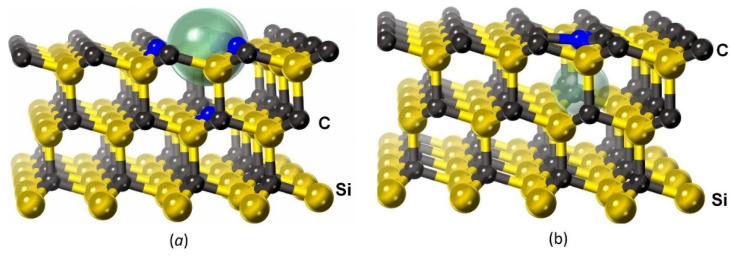
3C–SiC with the silicon vacancy $V_{Si}$ (**a**) or $C_{4}V$ center formed by a leap of the C atom from the bottom upward to the place of the vacancy (**b**). Semitransparent spheres denote voids; blue color marks C atoms having dangling bonds and a magnetic moment.

**Figure 7 materials-15-04653-f007:**
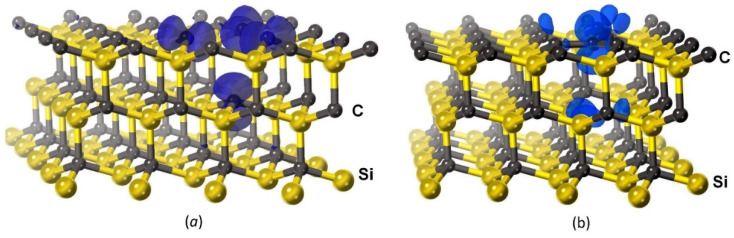
Difference in the density of electrons with spin “up” and “down”, corresponding to the value of $0.05\;e/{Å}^{3}$ for the systems with $V_{Si}$ (**a**) and $C_{4}V$ (**b**). Cases (**a**,**b**) are specific to SiC grown on Si of p- and n-type, respectively.

**Figure 8 materials-15-04653-f008:**
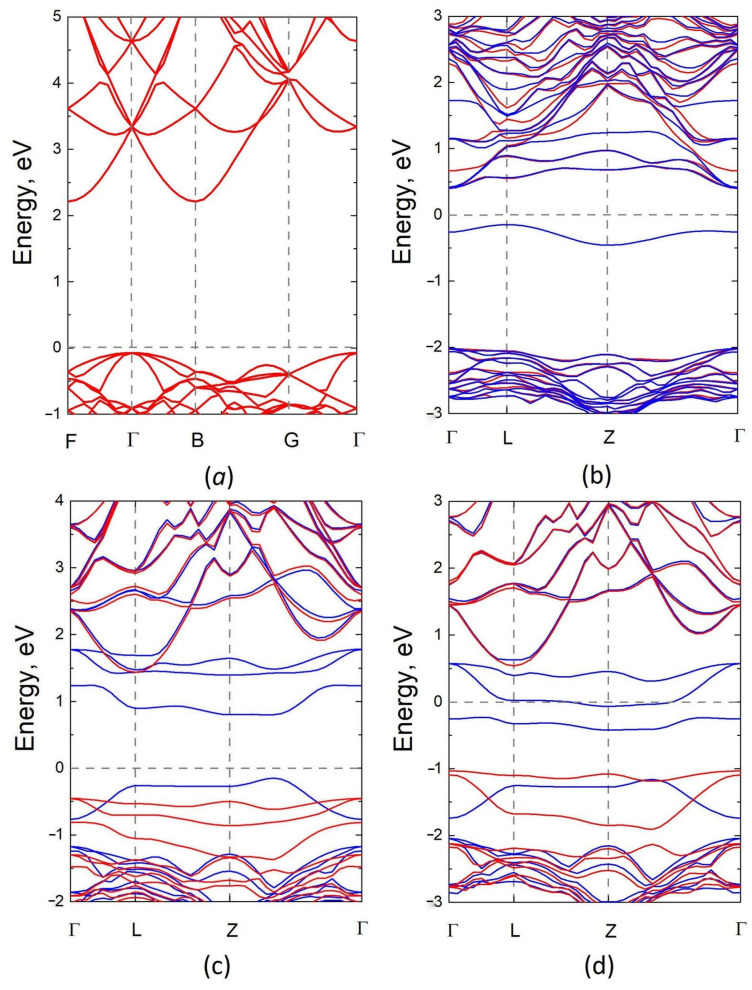
Band structure of an ideal 3C–SiC with the P1 symmetry and q=0 (**a**) and those of 3C–SiC containing silicon vacancies (P3m1 symmetry, $n_{V_{Si}}=3.7\;\%$ ) for charges q=+e (**b**), q=−e (**c**), and q=−2e (**d**). Cases (**b**,**c**) correspond to a magnetic semiconductor, while (**d**) is a magnetic semimetal. In plots (**b**–**d**), the bands for electrons with spin “up” and “down” are shown in red and blue, respectively, and there are new blue zones near the Fermi level (E=0 ).

**Figure 9 materials-15-04653-f009:**
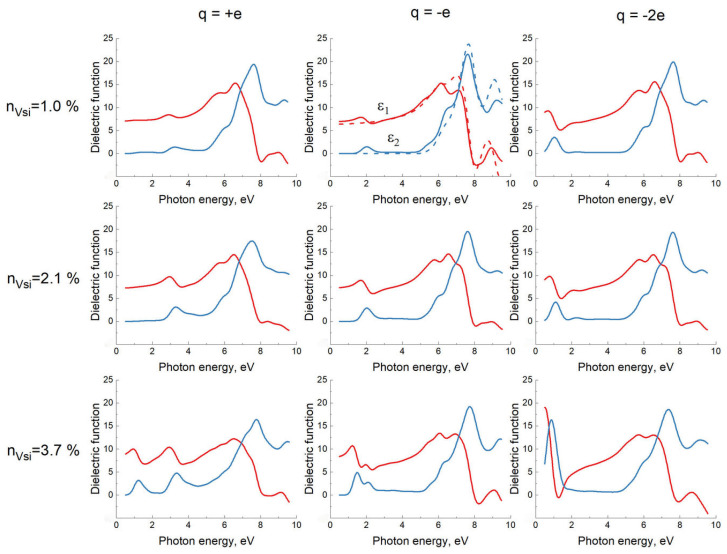
Dielectric function of 3C–SiC containing silicon vacancies with different concentrations $n_{V_{Si}}$ and different charges q, as calculated by the DFT method. The red color shows the real part $\varepsilon_{1}$ and the blue color shows the imaginary part $\varepsilon_{2}$. In the middle of the upper row, the dashed line shows the dielectric function of an ideal 3C–SiC.

**Table 1 materials-15-04653-t001:** Characteristics of the most favorable configurations of atoms at different values of vacancy charges *q*.

Parameter	The Case q=+1e	The Case q=0	The Case q=−1e	The Case q=−2e
$E_{C_{4}V},\;eV$	−2.6	−1.3	0.3	1.0
$\mu_{C_{4}V}$, $\mu_{B}$	1.0	1.0	0.5	0.0
$E_{M,\;\;C_{4}V},\;eV$	0.05	0.07	0.01	0.00
$\mu_{V_{Si}}$, $\mu_{B}$	1.0	2.0	3.0	2.0
$E_{M,V_{Si}},\;eV$	0.06	0.12	0.46	0.21

## Data Availability

Data are contained within the article.

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
