# Peer review of "Dielectric Function and Magnetic Moment of Silicon Carbide Containing Silicon Vacancies"

_materials, 2022, doi:10.3390/ma15134653_

Round 1
Reviewer 1 Report
For this manuscript, SiC is indeed an important material for electronics and epitaxy technique. I just have three small questions about this article.
1. The authors need to explain the shoulder peak at Si(111) position in XRD.
2. RHEED images should have some marks to show the structural information exactly.
3. All the symbol "~" have been written in "÷". Please revise them.
Author Response
Answers to the first reviewer
“Dielectric function and magnetic moment of silicon carbide containing silicon vacancies”
The authors express their deep gratitude to the reviewers for valuable comments that significantly improved the manuscript. The authors agree with all the comments in accordance with which this article has been corrected.
Reviewer #1:
- The authors need to explain the shoulder peak at Si(111) position in XRD.
This shoulder is an artifact that makes no sense. Thank you very much for pointing this out. We repeated the measurements to get rid of it and made new figure.
- RHEED images should have some marks to show the structural information exactly.
We have corrected Figure 5, which shows the RHEED images, and given the index numbers corresponding to the diffraction reflexes.
- All the symbol "~" have been written in "÷". Please revise them.
Corrections have been made accordingly.

Reviewer 2 Report
Lines 149 and 173. Authors determined the thickness of SiC layer in the samples by comparing experimental and theoretical dependences of ε(E). Does the authors checked the thickness of SiC by other technique (SEM for example) to support their calculations?
In my opinion, it would be useful to compare experimental ant theoretical results in the same graph, now it is not clear how big the mismatch between these results is.
Author Response
Answers to the second reviewer
“Dielectric function and magnetic moment of silicon carbide containing silicon vacancies”
The authors express their deep gratitude to the reviewers for valuable comments that significantly improved the manuscript. The authors agree with all the comments in accordance with which this article has been corrected.
Reviewer #2:
- Lines 149 and 173. Authors determined the thickness of SiC layer in the samples by comparing experimental and theoretical dependences of ε(E). Does the authors checked the thickness of SiC by other technique (SEM for example) to support their calculations?
Yes, we checked, and in both cases, the film thicknesses obtained by elipsometry coincided with those observed by scanning electron microscopy (SEM). So, we have inserted the corresponding text (Line 174) and the Figure 3 with SEM of both samples to prove it. Therefore, the numbering of the figures has changed.
- In my opinion, it would be useful to compare experimental ant theoretical results in the same graph, now it is not clear how big the mismatch between these results is.
Unfortunately, this is impossible, since this article deals only with a qualitative comparison. For quantitative comparison it is necessary to know the concentration of vacancies more precisely, and in addition, it is necessary to know their mutual arrangement, i.e., the symmetry group. This article is essentially the first publication on this topic. In the future, we will study the interaction between vacancies, find the most favorable symmetry group, and quantitatively compare the theoretical and experimental dielectric function.

Reviewer 3 Report
The article is devoted to the study of the properties of silicon carbide layers containing silicon vacancies, which were grown by the method of coordinated substitution of atoms. In general, the presented study is quite interesting and promising not only from a fundamental point of view, but also from a further practical application. The article corresponds to the declared journal and can be accepted for publication in the future after the authors answer a number of questions that have arisen during its analysis.
1. The abstract needs to be improved, the authors should reflect in more detail the novelty and practical significance of the work.
2. According to X-ray diffraction data, a large amount of silicon is observed in the structure. The authors should provide data from a detailed study of the phase composition, as well as changes in structural parameters depending on the type of silicon carbide obtained.
3. The authors should describe in more detail the mechanism of transformation of vacancy defects into silicon carbide nuclei during the formation of layers. It is also worth paying attention to the possibilities of forming these structures and their homogeneity.
4. The authors should pay more attention to the description of the formation of vacancy defects.
5. You should also pay attention to the strength properties of the resulting structures, how much do they differ from analogues?
6. Conclusion requires significant revision and optimization, as well as reflection of further research prospects.
Author Response
Answers to the third reviewer
“Dielectric function and magnetic moment of silicon carbide containing silicon vacancies”
The authors express their deep gratitude to the reviewers for valuable comments that significantly improved the manuscript. The authors agree with all the comments in accordance with which this article has been corrected.
Reviewer #3:
- The abstract needs to be improved, the authors should reflect in more detail the novelty and practical significance of the work.
The abstract has been revised to reflect in more detail the novelty and practical importance of the work.
- According to X-ray diffraction data, a large amount of silicon is observed in the structure. The authors should provide data from a detailed study of the phase composition, as well as changes in structural parameters depending on the type of silicon carbide obtained.
All information related to XRD has been reworked and Figures 4 and 5 have been replaced with new ones.
- The authors should describe in more detail the mechanism of transformation of vacancy defects into silicon carbide nuclei during the formation of layers. It is also worth paying attention to the possibilities of forming these structures and their homogeneity.
We added the following paragraph to our article on page 2 “In the review [10] a sequential thermodynamic and kinetic theory of Si transformation into SiC in the course of chemical reaction (1) was constructed. It was shown that the process of transformation of Si into SiC consists of two stages. In the first stage of the reaction (1), the CO molecule interacts with the surface of the silicon substrate. The oxygen atom reacts chemically with the Si atom to produce SiO gas. This reaction proceeds through the formation of an intermediate Si-O-C complex forming an almost equilateral triangle [14]. In fact, the oxygen atom plays the role of a reaction catalyst, without which the reaction (1) is impossible. Then the SiO gas is removed from the system, and a silicon vacancy is formed in place of the substrate silicon atom that turned into SiO gas. Since this is the first kind of phase transition, it occurs with the formation of a new phase nucleation. In this case, this nucleus is a complex formation consisting of the SiC itself, the pore under its surface and the surrounding shrinkage pores. An image of such a nucleus is given in the review [11]. For the thermodynamic and kinetic description of this kind of nucleation, the equations of the standard nucleation theory had to be significantly modified. The derivation and solution of these equations are given in the review [10]. In the same work, a theory of the formation of such nuclei on vicinal surfaces of Si crystals deviated by a small angle from planes (111) was also constructed.”
- The authors should pay more attention to the description of the formation of vacancy defects.
The same paragraph is a response to this comment.
- You should also pay attention to the strength properties of the resulting structures, how much do they differ from analogues?
This is a perfectly valid question. We investigated and continue to investigate as mechanical properties of films SiC on Si, and mechanical properties of bulk single crystals SiC These researches are published in series of works from which we result only three of them [1. Grashchenko A. S., Kukushkin S. A., and Osipov A. V. Nanoindentation and Deformation Properties of Nanoscale Silicon Carbide Films on Silicon Substrate // Technical Physics Letters. 2014. V. 40. â„– . 12. P. 1114-1116. doi: 10.1134/S1063785014120268; 2. S., Kukushkin S. A., and Osipov A. V. Study of Elastic Properties of SiC Films Synthesized on Si Substrates by the Method of Atomic Substitution // Physics of the Solid State. 2019. V. 61. â„–. 12. P. 2310-2312. doi: 10.1134/S106378341912014X.; 3. Osipov A. V., Grashchenko A. S., Gorlyak A. N., Lebedev A. O., Luchinin V. V., Markov A. V., Panov M. F., and Kukushkin S. A. Investigation of the Hardness and Young's Modulus in Thin Near-Surface Layers of Silicon Carbide from the Si- and C-Faces by Nanoindentation// Technical Physics Letters. 2020. V. 46. â„– 8. P. 763-766 doi: 10.1134/S106378502008012X. These studies showed that the top thin layers of SiC, about 17 nm thick have a microhardness equal to 43 GPa, which is about 30% higher than the microhardness of bulk silicon carbide. Then at a depth of 20-40 nm the hardness decreases sharply to a value equal to the microhardness of pure silicon. Young's modulus of SiC film determined in [Grashchenko A. S., Kukushkin S. A., and Osipov A. V. Nanoindentation and Deformation Properties of Nanoscale Silicon Carbide Films on Silicon Substrate // Technical Physics Letters. 2014. V. 40. â„– . 12. P. 1114-1116. doi: 10.1134/S1063785014120268] was equal to 328 GPa, which is only slightly lower than the Young modulus C of the face of bulk 4H-SiC crystal, which is equal to 400 GPa [Osipov A. V., Grashchenko A. S., Gorlyak A. N., Lebedev A. O., Luchinin V. V., Markov A. V., Panov M. F., and Kukushkin S. A. Investigation of the Hardness and Young's Modulus in Thin Near-Surface Layers of Silicon Carbide from the Si- and C-Faces by Nanoindentation// Technical Physics Letters. 2020. V. 46. â„– 8. P. 763-766 doi: 10.1134/S106378502008012X.] However, in these works the SiC on Si layers grown by the first method of growth, in which the preliminary annealing in vacuum was not carried out, were investigated. Studies of samples grown by the second method, in which preliminary annealing of samples in vacuum is carried out at the moment and, as yet, are not completed. We think that microhardness studies are not relevant to the topic of this article. This is an entirely different topic to which a separate paper will be devoted. In this article we did not touch it and did not give references to our works, which are presented above in the answer.
- Conclusion requires significant revision and optimization, as well as reflection of further research prospects.
We introduced an additional item in Conclusion. “Studies performed as a result of this work and earlier work [25] have shown that SiC grown on Si by MCSA method is a very promising material for use in spintronics and creation of a new type of electronic devices based on it. In this sense, silicon carbide obtained using the method of pre annealing Si in a vacuum is especially interesting, because in this case SiC is maximally saturated with silicon vacancies, which provide SiC magnetic moment. A study of the magnetic properties of SiC films on Si performed on SiC obtained by the MCSA method without silicon pre annealing [25] can serve as a proof of this. This SiC contains much lower vacancy density than the SiC obtained with the pre annealing. And, nevertheless, strong magnetic susceptibility was experimentally detected in the samples of this silicon carbide [25]. Obviously, in SiC samples with a high vacancy content this effect will only increase, which we hope will be damaged in the near future by our experiments.
Sincerely yours,
A. Kukushkin, A. V. Osipov,

Round 2
Reviewer 3 Report
The authors answered all the questions posed, the article can be accepted for publication.